# Repulsion leads to coupled dislocation motion and extended work hardening in bcc metals

K. Srivastava[1,2], D. Weygand [1✉], D. Caillard[3] & P. Gumbsch[1,4]

Work hardening in bcc single crystals at low homologous temperature shows a strong orientation-dependent hardening for high symmetry loading, which is not captured by classical dislocation density based models. We demonstrate here that the high activation barrier for screw dislocation glide motion in tungsten results in repulsive interactions between screw dislocations, and triggers dislocation motion at applied loading conditions where it is not expected. In situ transmission electron microscopy and atomistically informed discrete dislocation dynamics simulations confirm coupled dislocation motion and vanishing obstacle strength for repulsive screw dislocations, compatible with the kink pair mechanism of dislocation motion in the thermally activated (low temperature) regime. We implement this additional contribution to plastic strain in a modified crystal plasticity framework and show that it can explain the extended work hardening regime observed for [100] oriented tungsten single crystal. This may contribute to better understanding the increase in ductility of highly deformed bcc metals.

[1] Institute for Applied Materials (IAM), Karlsruhe Institute of Technology (KIT), Straße am Forum 7, 76131 Karlsruhe, Germany. [2] Research and Development, AG der Dillinger Hüttenwerke, Werkstraβe 1, 66763 Dillingen/Saar, Germany. [3] CEMES-CNRS, 29 rue Jeanne Marvig, BP4347, F-31055 Toulouse Cedex 4, France. [4] Fraunhofer IWM, Wöhlerstr. 11, 79108 Freiburg, Germany. ✉email: Daniel.Weygand@kit.edu

Metallic materials are used mainly in technical applications for their good formability, strength and toughness. This favourable combination of materials properties fundamentally relies on work hardening during plastic deformation. The response of metals to a mechanical load by irreversible plastic deformation occurs on a microscopic scale by the motion of dislocations, line defects of the crystalline structure. Dislocations in general glide on densely packed crystallographic planes. Dislocation glide on such glide planes causes a relative shift of the material above and below the glide plane in the direction of a short lattice vector, the Burgers vector[1]. The actual plastic deformation is a consequence of the interplay between dislocation glide, dislocation multiplication and annihilation on well-defined slip systems, characterised by glide plane and Burgers vector. Dislocations move collectively upon straining the material and multiply. Dislocation multiplication leads to an increase in dislocation density. This in turn is believed to result in work hardening which manifests itself as an increase in the flow stress of the material upon straining and gives e.g. a deep-drawn component its strength but also hinders further deformation. A suitable heat treatment can often restore formability.

Work hardening is known to depend on the crystalline structure of the metal, e.g. face centred cubic (fcc) metals like aluminium or copper typically show stronger hardening than body centred cubic (bcc) metals like iron or tungsten[2–5]. The long-range interaction between dislocations on different glide planes and with different character is caused by the elastic distortion that dislocations introduce into the crystal lattice. Additionally, short-range interaction at the intersection of dislocations can lead to changes in the atomic configuration of the dislocation core and to the formation of so-called dislocation junctions[2,3,6–8]. The prominent interpretation of work hardening as forest hardening is based on such interaction of dislocations on inclined slip systems hindering the motion of the mobile dislocations and thus leading to an increase in the stress for further plastic deformation[2,8,9]. Dislocation interactions may also result in multi-junctions involving more than two dislocations which in bcc metals have been shown to contribute significantly to work hardening[10].

All work hardening models intrinsically assume that the mutual interaction between dislocations always hinders dislocation motion either by repulsion on approach or by pinning at junctions, which prohibits the release and further motion of the dislocation. Therefore work hardening models all include a contribution to the flow stress which is inversely proportional to the average dislocation spacing $L$ which in turn is related to the dislocation density by $L \approx 1/\sqrt{\rho}$[4,8].

While these implicit assumptions and the resulting work hardening models appear plausible for fcc metals or bcc at high temperatures $T > T_c$ ($T_c$ is the athermal temperature) where dislocation glide is controlled only by the resolved shear stress on the glide plane, it is highly speculative for bcc metals at $T < T_c$ whose deformation behaviour is known to be much more complex[5,11–13]. A little understood example is the work hardening of tungsten single crystals at room temperature[14,15]. At room temperature, for [100] tensile loading, an extended work-hardening regime is observed with an initial flow stress of about 250 MPa and with a work hardening that leads to flow stresses of almost 1 GPa during straining by 6–8%[14]. Also a recent atomistically informed crystal plasticity model does not capture the initial hardening behaviour[9].

In bcc metals at low temperatures screw and non-screw dislocations have a completely different response to the applied stress $\sigma_{app}$. Non-screw dislocations can bend and their motion is controlled exclusively by the resolved shear stress $\tau_{res} = m\sigma_{app}$ on the glide plane, where $m$ is the so-called Schmid factor. In contrast screw dislocations remain straight and additional components of the stress tensor influence their glide behaviour[16,17]. The complex core structure of screw dislocations and its sensitivity to non-glide stresses is at the origin of this so-called non-Schmid behaviour[16,17]. This core structure is believed to make screw dislocations glide by the successive nucleation and motion of pairs of kinks on an otherwise straight screw dislocation line and lead to the thermally activated deformation behaviour of bcc metals at low temperatures[18]. Atomistic simulations and first principle calculations[19–25] have been used to investigate the properties of screw dislocations and of kink-pair formation for bcc materials. The core structure is found to be non degenerate and compact. The complex screw dislocation core structure also leads to the activity of unexpected glide systems[26]. However, the role of forest dislocations on work hardening in the low-temperature regime is not understood.

Since screw dislocations determine the deformation behaviour at low temperatures, we systematically investigate the mutual interaction of screw dislocations in bcc metals using discrete dislocation dynamics (DDD) simulations and in situ experiments. Tungsten (W) is chosen as a representative of bcc metals. Tungsten is elastically nearly isotropic and its mechanical properties are well studied and show extended work hardening regimes for [111] and even more pronounced for [100] orientation[14,15,27–29]. The investigation allows to identify a hitherto overlooked mechanism based on the coupled motion of repulsive oriented screw dislocation pairs in both simulations and experiments. The mechanisms' origin is the internal shear stress increase leading to screw dislocation glide on unexpected slip systems and thus a larger plastic deformation at the same macroscopic load occurs. This is a key aspect explaining the orientation dependence of the extended work hardening regimes, e.g. in tungsten.

## Results

**Discrete dislocation dynamics simulations**. DDD simulations are used to investigate the interaction of two screw dislocations on non-coplanar glide systems. All possible types of interaction pairs are studied. An atomistically informed DDD code[24,30] is employed here. Screw dislocation mobility is modelled by the kink-pair formation mechanism[18,30], using atomistic values for the stress dependent activation enthalpy for kink-pair formation (details in Methods section) thus including also non-Schmid effect. While attractive interactions show the expected dislocation reactions[31,32] and act as obstacles by forming dislocation junctions, specific repulsively oriented screw dislocation pairs surprisingly show coupled gliding. Such coupled motion, where one mobile screw dislocation (dislocation I), upon interaction with a second repulsively oriented (immobile) screw dislocation (dislocation II), drives the second dislocation without slowing down. This coupled motion occurs at constant applied stress. For room temperature deformation, modelled here, the kink motion along the dislocation line (2 μm in length) occurs faster than the time scale of kink-pair nucleation as long as the kinks see a positive driving force.

Figure 1a displays a schematic diagram of the simulation setup: two repulsively oriented screw dislocations, dislocation I belonging to glide system $(\bar{1}01)[111]$ and dislocation II to glide system $(011)[11\bar{1}]$, are placed within a crystal subjected to tensile loading along the $[\bar{1}49]$ direction. The Schmid factors for dislocations I and II are $m_I \approx 0.5$ and $m_{II} \approx 0.3$. The nearest distance between the dislocations is initially about 50 nm. Dislocation I begins to glide once the resolved shear stress $\tau_{res\ I}$ reaches about 1.8 GPa. The corresponding resolved shear stress on dislocation II is about 1.2 GPa. The glide directions of the dislocations I and II are

marked as $r_1$ and $r_2$ and both are pointing to the left in Fig. 1a. Dislocation I reaches a velocity of about 1.05 nm/s, while dislocation II is initially immobile (see Fig. 1b).

Figure 1b shows the velocity of the two dislocations as a function of their nearest distance of approach. Dislocation I approaches the immobile dislocation II causing it to glide once the nearest distance decreases to about 9.5 nm. From that point on, the velocity of dislocation II increases rapidly to match the velocity of dislocation I. Their nearest distance then stabilises around a value of $d_{crit} \approx 7.3$ nm. Both dislocations then glide collectively.

The stress states along both dislocation lines favour glide on their initial habit planes. Both dislocations remain straight throughout the entire simulation. No cross slip is triggered (see Methods) due to the mutual interaction. The maximal resolved shear stresses along the dislocations due to their mutual elastic interaction at the point of nearest distance is shown in Fig. 2a. The distribution of the negative (positive) additional shear stress acting on dislocation I (dislocation II) along the dislocation line for the distance $d_{crit}$ is shown in Fig. 2b). The curves are asymmetric with respect to the nearest point of interaction as the respective resolved shear stresses are shown.

The coupled motion of the two dislocations can be understood based on this analysis of the stress distribution along the dislocation line. Over the entire length of dislocation I (about 2 μm) only a very short section of about 20 nm around the point of closest interaction experiences a significantly reduced resolved shear stress. Consequently, kink-pair nucleation is reduced very locally in this section, while all the rest of the dislocation line still experiences high resolved shear stresses from the externally applied field and kink-pair nucleation rate remains virtually unchanged. Therefore, the total velocity of dislocation I remains essentially constant. The situation of dislocation II is opposite: once dislocation I has reached the critical distance, the kink-pair nucleation rate on dislocation II drastically increases near the point of closest approach until the dislocation velocity reaches the velocity of dislocation I. Thereafter the two dislocations show coupled glide, slightly oscillating around the critical distance due to the different orientations of the elementary kink steps on the two dislocations.

**In situ transmission electron microscopy.** To mimic this scenario experimentally, repulsively oriented screw dislocations in

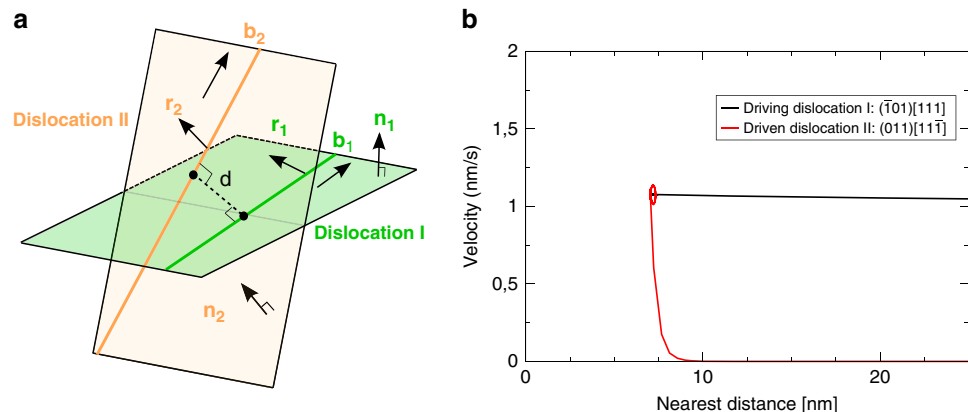

**Fig. 1 Simulation setup and coupled motion. a** Schematic view of the setup with two repulsively oriented screw dislocations I and II with Burgers vectors $b_1$ and $b_2$ respectively. The indicated directions $r_1$ and $r_2$ are parallel to the Peach–Kohler force due to the external loading resolved in the respective primary glide planes. Only repulsive pairs, where the glide directions $r_1$ and $r_2$ have a positive scalar product have to be considered. The location A and B marked on the dislocation line are the nearest points between the two dislocations defining the nearest distance d. **b** The variation of the velocity of dislocations I and II with Burgers vectors [111] and [11$\bar{1}$] is plotted versus the minimal distance of approach.

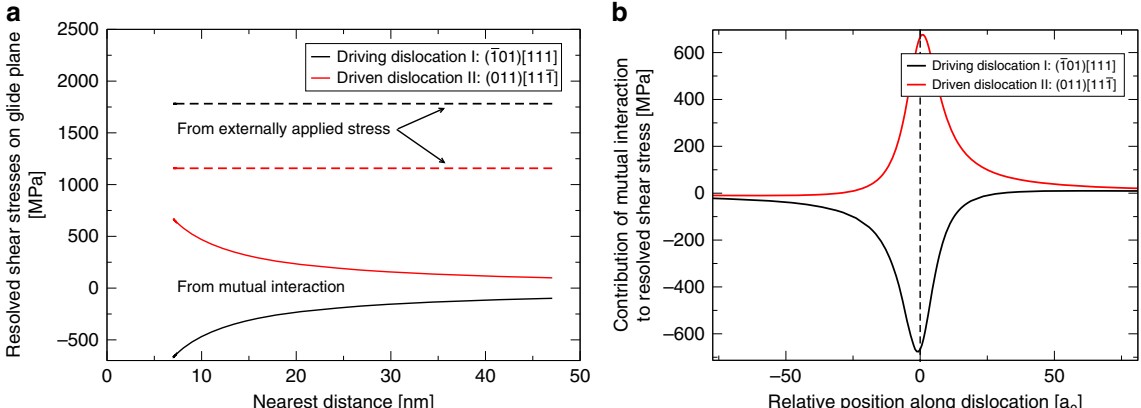

**Fig. 2 Stress distribution on repulsively interacting dislocations. a** Resolved shear stress from externally applied stress on dislocation I and II (dashed line) and shear stress due to mutual dislocation interaction (line) at the closest point of interaction (nearest distance); **b** resolved shear stress due to the repulsive dislocation interaction along dislocation I and II. Dislocations are at distance $d_{crit}$. The relative positions are given in multiples of the lattice constant $a_0$ and measured from the closest point of interaction.

thin films of a tungsten singe crystal are investigated by TEM. Thin film specimens of tungsten suitable for in situ deformation experiments have been prepared and strained in a transmission electron microscope (TEM, see Methods for a description of the method and Caillard[13] and Supplementary Figs. 1, 2 for corresponding observations in Fe and Nb). The foil plane is $(\bar{1}\bar{1}7)$, the local tensile axis is [521], and the sample thickness is 300 nm. Under such conditions, repulsively oriented dislocation pairs could be identified: Dislocations of type I with Burgers vector $1/2[\bar{1}\bar{1}1]$ and length 360 nm are the most mobile in the (101) plane, in agreement with a high Schmid factor of 0.485. Dislocations of type II with Burgers vector ½[111] later move on the $(\bar{1}01)$ plane, but initially are immobile due to the lower Schmid factor of 0.458.

Figure 3i shows the schematic of the TEM sample geometry. The slip planes of the studied dislocations intersect the two free surfaces along the directions noted slip traces, and intersect each other along the direction noted node path. Dislocations are represented by the straight lines marked I and II, gliding to the left on the respective slip planes 1 and 2. Figure 3a–g shows the TEM observation of these dislocations for different times during the deformation at $T = 300\,K$ (see also the Supplementary Movie). In the TEM sample, the first respectively second driving dislocation is labelled dislocation Ia respectively dislocation Ib. Dislocation Ia approaches dislocation II between (a) and (b). Note that dislocation II is immobile with respect to the fixed point 'x'. Coupled motion takes place in (c) and (d) (see fixed points 'x' and 'y') until (e) where dislocation Ia has moved away

leaving dislocation II again immobile with respect to the fixed point 'z', between (e) and (f). The velocity of dislocation Ia remains almost constant close to 25 nm/s during the whole process. Then a second dislocation Ib arrives and pushes again dislocation II in (f) and (g). The image (h) which is the difference between (a) and (d) shows the starting positions in dark and the final ones in bright. The velocity of dislocation II fluctuates around the one of the driving dislocation Ia or Ib. The fluctuations are more pronounced than for the DDD results.

During this coupled motion, the point of closest distance moves along the direction [010] of intersection between the two slip planes. Since this direction is almost within the plane of the TEM-foil, long distances of coupled motion can be observed. The minimum distance during the coupled motion has been measured in projection, and corrected from perspective effects. At 300 K, this distance fluctuates a little bit, but remains equal to about 24 nm when coupled motion takes place. Similar observations of collective glide have been made a $T = 200\,K$, but the corresponding critical distance is unfortunately too small to be measured.

## Discussion
Both experiments and simulation show coupled motion of repulsively oriented screw dislocation pairs. In both cases screw dislocations remain straight and no cross slip is observed. The DDD simulations suggest that this behaviour is a direct consequence of dislocation motion by the kink-pair mechanism. Kink-pair nucleation occurs locally and therefore depends on the

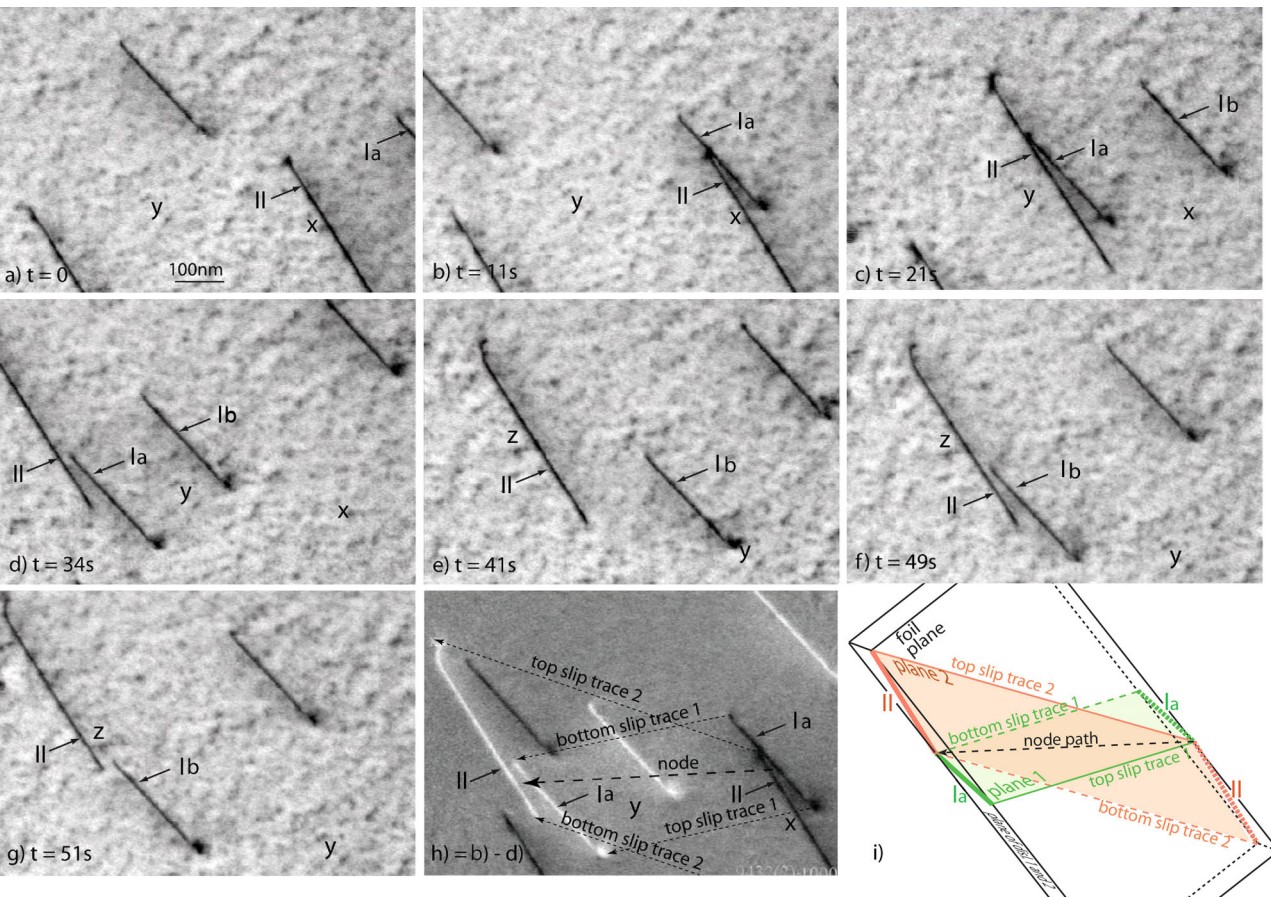

**Fig. 3 In situ TEM observation of coupled glide: experimental observation of dislocation glide at different time steps.** Reference points are indicated by 'x, y, z'. **a** Dislocation Ia approaches the still dislocation II. **b–d** Dislocation Ia pushes dislocation II: their point of closest distance moves along the direction 'node path' until it emerges at the bottom surface in (**e**). **f, g** Another dislocation Ib pushes dislocation II. **h** Difference-image showing the motion between (**b**) (black dislocations) and (**d**) (bright dislocations). **i** Scheme corresponding to (**h**). The video sequence can be downloaded as a Supplementary Movie.

stress state along the entire screw dislocation line[18,30]. If the critical resolved shear-stress (CRSS) is reached upon raising the applied stress, dislocations of type I move while dislocations of type II should and do remain initially immobile in experiment and simulation. Once the two dislocations get close enough, the region of closest encounter on dislocation II experiences resolved shear stresses which are higher than the stresses on dislocation I. This has the consequence that kink pairs on dislocation II are generated in this region of closest encounter at a rate (per unit length) somewhat higher than on dislocation I, where nucleation occurs along the entire line. The kinks generated at this location of closest encounter move along the dislocation line as long as there is a positive driving force which is true for both experiments and simulation. Dislocation I could in principle slow down slightly because part of the total length of the dislocation along which kink-pair nucleation may occur is now at lower stresses (c.f. Fig. 2b). However, already at our line length of 2 μm, this slowing down is too small to be observed in the simulations. Eventually the total kink generation rates along both dislocations match and both dislocations move at the same velocity. This implies that these repulsively oriented screw dislocations should never cross each other as long as both are able to move. Motion of a primary screw dislocation can activate certain repulsively oriented screw dislocation (detailed in Supplementary Discussion) even if it is located on a slip plane with a very low resolved shear stress.

Even quantitatively, the detailed TEM investigations and the simulations can well be related to each other. The smaller differences in Schmid factor in the experimental setup compared to the simulations make dislocation II move somewhat earlier, i.e. at an about 3 times larger critical distance, than in the simulations. The TEM investigation also shows clearly that the mechanism is repeatable for dislocation II as dislocation Ib moving on a glide plane parallel to that of dislocation Ia takes over the role of the driving dislocation as dislocation Ia moves out of the observed area of the thin film. During in situ experiments at lower temperature[13] much shorter critical distances between the coupled dislocations are observed which are beyond the limit of resolution. The smaller critical distance between the coupled pairs leads to larger interaction stresses, as required for activating screw dislocation motion at lower temperatures.

Since our observations can be explained by dislocation motion based on kink-pair nucleation alone, this coupling mechanism between two repulsively oriented screw dislocations is likely to be observed in all bcc metals and other materials where kink-pair generation governs dislocation motion. In Supplementary Figs. 1, 2 additional TEM evidence for the coupling in Fe and Nb at $T =$ 95 K is presented, confirming the universality of the phenomenon.

The in situ observations and DDD results are both obtained for straight dislocations, reaching from surface to surface. In bulk material, dislocations are often anchored in networks which leads to an increase in line length during motion. One could speculate that this might possibly suppress screw dislocation motion and hence the coupling mechanism. However, the increase of length corresponds to the nucleation of new kink pairs, which takes place anyway, even in case of straight screws ending at the surfaces. These kinks subsequently either accumulate at the dislocation extremities in the case of bulk material or disappear at the surfaces. This increase of length is actually a restoring force, but already included in the kink-pair nucleation process. Therefore, the only ingredient, which has not been taken into account for mimicking bulk behaviour, is the elastic repulsion between kinks accumulated at the screw extremities—forming a mixed dislocation (curved section with non-screw character). This repulsion is part of the internal stress due to elastic interactions with all neighbouring dislocations, which of course must be taken

into account for kink-pair nucleation in bulk. However, this is very different from the restoring forces due to line tension effects observed in fcc metals.

The observed collective glide of dislocations has significant consequences on the evolution of the plastic strain rate in bcc metals in the thermally activated regime. Classical phenomenological work hardening relations have been proposed with a flow stress inversely proportional to the average dislocation spacing $L$ which can be calculated from the dislocation density as $\approx 1/\sqrt{\rho}$. Dislocations are considered there as mutual obstacles which leads to so-called forest hardening. Refined models consider an effective hardening between slip systems by specific hardening coefficients[32–34].

The situation for bcc metals in the thermally activated regime differs: the screw dislocations in the repulsive pairs remain straight. Our results show that the high effective 'stiffness' of screw dislocations leads to a coupled glide of dislocation ensembles, here exemplarily studies at pairs or groups of screw dislocations (see Supplementary Note 1 and Supplementary Fig. 5). Within the range of validity of the kink-pair nucleation mechanism for dislocation motion, the mutual interaction of these repulsively oriented dislocation pairs is limited to modifying the local stress state in the zone of closest approach. Therefore it locally enhances the kink-pair nucleation rate.

During coupled motion the second dislocation (slip system 2) contributes to the total plastic shear rate on slip system 2, due to dislocation glide on slip system 1, even though macroscopically the resolved shear stress is below its CRSS. In order to take such a phenomenon into account we propose to modify the equations for the plastic shear rate evolution by adding an interaction term, expressing this coupling for the well accepted strain rate based formulation[18,35]

$$\dot{\gamma}_{\text{std}}^{\alpha} = \dot{\gamma}_0 \exp\left(-\frac{\Delta H_\alpha(\sigma)}{k_{\text{B}} T}\right), \tag{1}$$

where $\dot{\gamma}_{\text{std}}^{\alpha}$ denoted the standard plastic shear strain rate normally used in crystal plasticity (CP) formulations[36,37], $\Delta H_\alpha (\sigma)$ is the stress dependent activation enthalpy for system $\alpha$, $k_{\text{B}}$ is the Boltzmann constant and $T$ is the temperature.

The coupled motion of specific screw dislocation pairs is captured by adding a shear strain rate $\dot{\gamma}_{\text{coupl}}^{\alpha}$

$$\dot{\gamma}_{\text{tot}}^{\alpha} = \dot{\gamma}_{\text{std}}^{\alpha} + \dot{\gamma}_{\text{coupl}}^{\alpha} = \dot{\gamma}_{\text{std}}^{\alpha} + f_{\text{coupl}} \sum_{\beta \neq \alpha} K_{\alpha\beta} \dot{\gamma}_{\text{std}}^{\beta}, \tag{2}$$

where $\dot{\gamma}_{\text{tot}}^{\alpha}$ is then the total plastic shear strain rate. The coupling matrix $K_{\alpha\beta}$ parametrises the coupled motion between slip system $\alpha$ and $\beta$ and $f_{\text{coupl}}$ is a scaling factor between 0 and 1. The terms in the matrix $K_{\alpha\beta}$ depend on the crystallographic orientation of the two slip systems and the local stress state as several conditions (see also Supplementary Note 2) have to be fulfilled. The conditions are (i) repulsive orientation; (ii) the resolved Peach–Koehler force due to the stress state on both dislocations has a positive scalar product and (iii) include only repulsive dislocation pairs which can glide over extended distances (see also Supplementary Fig. 6).

In this analysis, only {110} slip systems are considered (Supplementary Table 1), as suggested by atomistic simulations[19]. In experimental observations for [110] loading slip activity on {112} systems is reported too[13]. Therefore the focus in the following is on [100] and [111] loading directions, where {110} slip planes are reported.

This coupling has important consequences on the plastic flow: for a given stress state, the coupling enhances the plastic strain without requiring an additional applied stress and thus the strain-hardening rate $\partial\sigma/\partial\epsilon$ is effectively lowered.

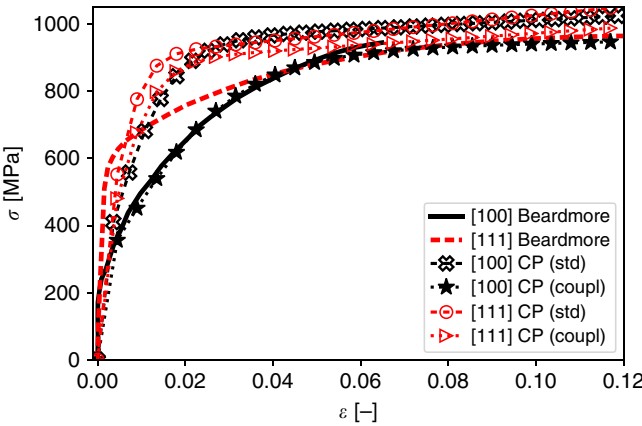

**Fig. 4 Deformation behaviour of tungsten single crystals for the [100] and [111] loading direction.** The experimental curves are taken from Beardmore and Hull[14]. For the [100] direction an extended work-hardening regime is observed; crystal plasticity simulation results without (CP(std)) and including the coupling mechanism (CP (coup)) are shown for both orientations. (A detailed comparison is included in Supplementary Discussion). The coupling mechanism leads to an extended work hardening regime for [100] representing well the experimental findings, while for the [111] orientation the deformation behaviour is only slightly changed.

The coupling mechanism is a key to rationalise the extended work hardening regime of tungsten single crystals at low homologous temperature for [100] loading direction shown in Fig. 4. Beardmore and Hull[14] and Argon and Maloof[15] observed for the [100] loading direction a gradual 'roundish' strain-hardening curve in contrast to [111] which display a classical hardening behaviour after yielding.

For the [100] loading direction a total of 12 coupled pairs exist, where both dislocations have the same non-zero Schmid factor. For the [111] only three coupled pairs exist. Details on the pair counting are given in Supplementary Note 2 and Supplementary Tables 2–5. Thus, the additional plastic strain at a given stress level due to the coupled pairs is most pronounced for loading along [100] direction. The value of $f_{coupl}$ was set to 0.5. Physically a value of 1 would mean, that each driving dislocation triggers a 'repulsive' event. For the chosen value of 0.5 every other dislocation triggers such an event. Furthermore, in case of [100] loading direction, a closed chain of coupled pairs is available, which further increases the plastic strain rate. The inclusion of the mechanism in the CP model (Supplementary Discussion and Supplementary Fig. 7) has a rather modest effect on the stress strain curve from the CP model for [111] loading. For the [100] loading direction, the coupling changes the initial hardening (CP (std) curve) quite drastically and leads to an excellent agreement with the experimental results (Fig. 4; stress strain curves marked by CP(coupl)).

In conclusion, a new mechanism influencing the work hardening in bcc metals in the kink-pair regime, based on the repulsive interactions between non-coplanar slip systems in bcc metals, is presented. The repulsively oriented forest screws, rather than obstructing the motion of an incoming screw dislocation, glide collectively at no additional applied stress. The present work shows that the interaction zone is limited to a very small region of the interacting dislocations. These results deeply change the concept of internal stresses, at least in bcc metals or materials where the kink-pair mechanism govern dislocation motion. In 2-D models it is generally assumed that moving dislocations are subjected to a uniform stress along their line, and that this stress oscillates in the direction of motion. Under such conditions, the average dislocation velocity is determined by the highest internal

stress $\sigma_{int}$ opposing to the applied stress, namely by a stress $\sigma - \sigma_{int}$. Here we show that in a 3-D model the velocity of dislocations is determined by the highest internal stress acting in the direction of the applied stress along the dislocation line, namely by a stress $\sigma + \sigma_{int}$. This strongly increases the average dislocation velocity in comparison with classical models, not only for the driven dislocations studied in this article, but also for all dislocations. We also obtain the surprising result that the dislocation velocity tends to increase with increasing internal stress, i.e. with increasing deformation. Depending on the loading state, the number of active repulsive dislocation pairs varies and is highest for the [100] orientation, explaining the observed extended work hardening behaviour of tungsten.

This strain-softening effect compensates for other classical strain-hardening ones and may also accounts for a lower global strain-hardening of bcc metals. These effects must be taken into account in macroscopic models like CP. The relevance for the coupled motion increases with dislocation density, as the density of possible pairs increases too. For very high dislocation densities, the internal stresses fluctuate over short distances, thus triggering possibly the coupled motion. Therefore this may also lead to the enhanced ductility observed for heavily cold rolled tungsten[38–41].

## Methods

**Discrete dislocation dynamics model and setup.** The following setup for the pillar is taken: the size of the box is 2 µm with an aspect ratio of 1:2:1. The DDD model used to simulate the screw dislocation motion is described in Srivastava et al.[30,42]. The mobility of the screw dislocation is governed by an Arrhenius law and accounts for the influence of the entire stress tensor on the activation energy of screw dislocations[30] as against a pure shear stress based formulation in literature[43–45]. The model effectively takes into account changes in the dislocation core structure due to the applied stress by including non-Schmid terms in the activation enthalpy. In the DDD model the effective kink-pair nucleation rate for the screw dislocation section is calculated on all three possible glide planes for a screw dislocation. In principle, a screw dislocation may glide on different glide planes and therefore split into distinct sectors depending on the local stress state. This is not the case for the current setups. For all non-screw orientations phonon drag limited glide is assumed as for fcc metals.

**Details to the repulsive pairs in the DDD setup.** The dislocations are labelled as I and II depending on their respective role. Dislocation I is mobile due to the externally applied loading and drives dislocation II, once a critical minimal distance is reached.

The dislocation I belongs to the slip system $(\mathbf{n}_I, \mathbf{b}_I) : (\bar{1}01)[111]$ and dislocation II belongs to the slip system $(\mathbf{n}_{II}, \mathbf{b}_{II}) : (011)[11\bar{1}]$. Both dislocations have screw orientation. The glide planes intersect along parallel to the $[1\bar{1}1]$ direction and the direction of shortest distance between the two screw dislocations is along $[1\bar{1}0]$ direction. In order to be repulsively oriented, the line direction of dislocation I is chosen parallel to its Burgers vector, while for dislocation II the line direction is antiparallel to its Burgers vector.

**Details on the local dislocation loading.** A uniaxial loading on the sample is applied, until dislocation I reaches a velocity of about 1 nm s⁻¹. During further simulation the externally applied load is kept constant. The Schmid factors on dislocation I respectively II are 0.5, respectively, 0.3.

The minimum activation enthalpy of kink-pair nucleation of dislocation II occurs at the position of nearest approach where the total interaction is strongest. The velocity of dislocation II is dominated by the kink pairs nucleated in the interaction zone which then spread out along the dislocation line. As mentioned in the main part, the overall direction of the Peach–Koehler force drive both the dislocation in a similar direction. Therefore once, kink pairs are nucleated, the stress level on the dislocation line outside the interaction zone drives these pairs along the dislocation line. On dislocation II the peak resolved shear stress exceeds the one of dislocation I by about 50 MPa leading to the same effective velocity required for coupled motion. This used velocity law implicitly assumes also that kink collision is unlikely[18,30,24]. For dislocation I this is obviously true, as the zone of interaction, where the kink-pair nucleation rate is drastically increased, is extremely small and both kinks will glide easily to the opposite sides of this zone. Outside this zone of interaction (nearest approach), kink-pair nucleation is very unlikely and therefore this assumption justified. For dislocation I the length dependency of the mobility law is supported by observations on Fe at room temperature[46].

**Stresses acting on the dislocations**. Coupled motion occurs only if both dislocations remain in their initial habit plane, therefore the question of cross slip has to be addressed: cross slip occurs only if the activation energy of glide for screw dislocation is minimum on a plane other than the habit plane. Due to symmetry in bcc metals, this limits the angle $\chi$ between the habit plane and the maximum resolved shear stress plane (MRSSP) to be within $-30° \leq \chi \leq 30°$. Supplementary Fig. 3a shows that the MRSSP angle $\chi$ of both screw dislocation remains within this range and therefore the lowest activation enthalpy is on the corresponding glide plane of the dislocation.

Supplementary Figure 3b shows that the corresponding activation enthalpy for the driven dislocation II in the interaction zone is reduced significantly below the activation enthalpy of the driving dislocation I because it needs to nucleate all its kinks in the short interaction zone.

**Screw dislocation velocity**. The local screw dislocation velocity is given by:

$$v = \frac{ba_0 L}{l_c^2} \nu_D \exp\left(-\frac{\Delta H(\sigma)}{k_B T}\right), \tag{3}$$

where $\nu_D$ is the Debye frequency, $b$ magnitude of the Burgers vector, $a_0$ is kink height on 110 systems ($a_0 = a\sqrt{2/3}$) and $a$ is the lattice constant.

$L$ is the screw dislocation length and $l_c$ is the critical length for the nucleation of the kink-pair, which is calculated from the activation volume as $l_c \approx \frac{V}{a_0 b}$[18].

**Experimental setup**. In situ straining experiments were carried out in a JEOL 2010HC transmission electron microscope working at 200 kV, using the room-temperature and the low-temperature straining holder designed by GATAN. The dynamic sequences were recorded by a Megaview III camera at the speed of 25 images/s, and analysed frame by frame. Rectangular microsamples were cut in a single crystal of high-purity tungsten described in[47]. They were mechanically thinned down to 10 μm thick and subsequently electro-polished with a NaOH solution until obtaining a thin edged hole at their centre. Then, they were glued on a copper grid fixed on the holder. The Burgers vectors were determined by the classical extinction rules using several diffraction conditions, and the slip planes were deduced from the directions and separation distances of the slip traces left by the moving dislocations at the two surfaces.

The local direction of the tensile axis can slightly deviate from the imposed one by several degrees in the foil plane. However, it can be determined with a pretty good accuracy in samples with rounded holes and containing no cracks, on the basis of finite element calculations. Local Schmid factors can then be determined with an accuracy of a few percent.

The local shear-stress intensity can be deduced from the critical widths of expanding screw dipoles, using elasticity models. Supplementary Figure 4 shows a screw dipole formed at 300 K in less than one frame of the video. After determining the dipole plane on the basis of the slip trace direction at its emergence point (noted tr.), the critical width corrected from perspective effects has been estimated as 85 nm in (c). Similar measurements on about ten expanding dipoles yield values comprised between 65 and 90 nm, corresponding to a local stress of 640 ± 100 MPa, at 300 K. This stress value is consistent with macroscopic values at 10% strain, or measured by the so-called incremental-straining-temperature-lowering tests, namely after the exhaustion of the mobile non-screw segments[27].

Stress measurements during in situ relaxation tests show a 14% decrease of local stress $\tau$ when the velocity $v$ of dislocations decreases by a factor of 7, which corresponds to a stress-velocity dependence $\frac{\Delta \ln v}{\Delta \ln \tau} \approx 15$.

TEM images are showing the screw dislocation coupling phenomenon in Fe and Nb at $T = 95$ K are shown in Supplementary Figs. 1, 2. The observations confirm that the coupling is a mechanism present in crystal structures, where (screw) dislocation have to overcome a barrier by the kink-pair mechanism in order to glide.

## Data availability

The datasets generated during or analyzed during the current study are available from the corresponding author on reasonable request.

## Code availability

Access to the code will be provided at the host institution of the corresponding author upon reasonable request. The occurrence of the coupling mechanism does not depend on code details. The basic principles are described under methods.

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

## Acknowledgements

P.G. acknowledges financial support from German Research Foundation (DFG) through project grant Gu367/30. D.W. acknowledges financial support for the research group FOR1650 Dislocation based plasticity funded by the German Research Foundation (DFG) under contract numbers WE3544/5-2. We acknowledge support by the KIT-Publication Fund of the Karlsruhe Institute of Technology.

## Author contributions

K.S. and D.W. conceived the simulation tools and conducted the simulations and analyzed and interpreted the data; wrote and edited the paper. D.C. conceived, designed, conducted and analyzed the experiments; wrote and edited the paper. P.G. provided the sample; contributed to the conception of the work; interpretation of the data, wrote and edited the paper.

## Funding

## Competing interests

The authors declare no competing interests.
