## [Peer Review File · Nature Communications]

Reviewers' Comments:

Reviewer #1:

Remarks to the Author:

The manuscript reports on a comparison between dislocation dynamics simulations and in-situ transmission electron microscopy regarding the repulsive interaction between two non-parallel screw dislocations. It is shown that the dislocations can glide together, producing an extra plastic deformation than can be linked to a lower strain hardening rate observed during deformation of bcc samples.

The work is potentially interesting although this Referee is concerned with the effect of the boundary conditions. In both DD and TEM, the dislocation ends are free, which allow the dislocations to glide over large distances. However, in a bulk sample, the dislocation ends would be part of a network and would be fixed, impeding the motion of the dislocations and limiting the production of plastic deformation. The same would be true in FCC metals: in absence of constraints on the dislocation ends, two repulsive dislocations can glide together and produce plastic strain. But this does not occur because dislocation ends are part of a network and cannot glide freely. Because of the nodes, the dislocation cannot remain straight and cannot maintain a constant velocity, in contrast with the present simulations and statements on P. 7, 12.

Other more minor concerns and comments:

- P.2 end of 1st paragraph: the referee does not understand « gives a formed component its strength » Isn't there a problem of English here?
- P. 3 2nd paragraph: at which temperature were the [110] tractions performed?
- P. 4: the referee does not agree that dislocation cores in bcc metals are extended and non-planar. It is quite the opposite. As shown by DFT calculations, non-degenerate core are highly compact. Having a compact core (and not an extended or non-planar core) is consistent with a high Peierls stress since it is known from the Peierls Nabarro model that compactness increases the Peierls stress. The authors should revise the corresponding paragraph
- One of the authors has published in-situ observations in tungsten with very surprising dislocation behaviors. Has any of this peculiarities of tungsten been accounted for in the present study?

Reviewer #2:

Remarks to the Author:

Authors report on a novel mechanism of dislocation interaction that acts contrary to conventional understandings of dislocation hardening in other metals. The observation is specifically recorded for tungsten-based metals, but shows broad generality to other BCC metals and systems that experience dislocation motion that relies on kink-pair nucleation processes. This particular observation is useful in understanding enhanced ductility in heavily cold-worked BCC metals at temperatures that are generally considered to be below the ductile to brittle transition; these specific issues have not been adequately described in the literature to date.

The author's claims are well supported by experimental transmission electron microscopy (TEM) observations collected during time-resolved straining experiments, and modelling efforts support these findings with atomistically-informed discrete dislocation dynamics (DDD) simulations. Each of these methods has fundamental physical limitations with respect to time resolution and appropriate boundary conditions; however, the authors adequately address these concerns and provide a compelling argument for a new mechanism governing the coordinated motion of dislocations in this material.

There are a couple of minor issues that should be discussed prior to publication. First, why are the orientations between dislocations and loading directions (and the resulting Schmid factors) for driving and driven dislocations different between TEM and DDD modeling experiments? Second,

why is the crystal plasticity modelling compared to the experimental results of Beardmore and Hull (1965) instead of the more recent publications by Brunner (2008, 2010), especially when the TEM experiments seem to be investigating material sourced from these latter investigations?

Overall, the manuscript is of very high quality with strong and conclusive support of the author's claims. The impact of this research is expected to be broad and substantial within the field of materials science. Furthermore, the methods employed may be helpful for elucidating mechanisms of deformation behavior in a variety of novel materials that are used in extreme and demanding engineering environments.

Sincerely,

Brady G. Butler, PhD

P.S. Minor issues with figures and formatting:

Figure 1a) Difficult to tell orientation of n_2 . It appears to be orthogonal to n_1 instead of normal to the plane containing Dislocation II. Figure 1 b) It would be helpful to clarify initial vs final configurations of dislocations and state that the critical distance is a function of the difference in Schmid factors (note: this is explicitly stated in a latter part of the manuscript). Also, what is the red circle at the intersection between dislocation 1 and 2? It looks like a continuation of the line describing dislocation II, but this mark and the truncation of the curves beyond the critical distance is not addressed.

Figure S4.1) This figure appears to be unformatted, making it difficult to tell orientation of different vectors. I would recommend a 3D view of this figure in the final supplemental section to improve clarity. Also, the legend uses shorthand text instead of the specific variables identified in the supplemental manuscript.

Reviewer #3:

Remarks to the Author:

This work uses a combination of multiscale modeling (DD and CP) with TEM observations of screw dislocations in W to evidence a new type of dislocation-dislocation interaction and its impact on work hardening in bcc metals. The results are new and of interest to the dislocation plasticity community as well as to a broader audience in the field of metallurgy and mechanical properties of materials.

Point by point comments:

- The title suggests the study concerns all bcc metals but the study was performed in W only. The reviewer agrees that the mechanisms evidenced here are most likely transferable to other bcc metals, but the justification for the use of W is not clear.
- The abstract does not follow standard rules (it starts with a result for example) and is too technical for a broad audience. The introduction however is very pedagogic so the reviewer believes the authors can improve the abstract.
- The text mentions several times that the DD and CP models are atomistically informed but remains vague as to how: which atomistic models are used as input, what phenomenon are taken into account (the fact that non-Schmid effects are accounted for is not mentioned in the main text, only in the Supplementary files), and how that may influence the results obtained here, especially the Schmid factors:
- The parameters given in S5 for the CP calculations do not include T/AT asymmetry ($a_1 = 0$)
- The authors find a different value of distance between gliding dislocations with DD and with TEM

and link it to the different Schmid factors. Could a quantitative link be made there? Were non-Schmid effects taken into account when calculating the Schmid factors with DD?

- Can non-Schmid effect be a source of discrepancy between the results from CP and experiments for [111] loading in Fig.4?
- Why was $[\bar{1}49]$ chosen for the tensile loading axis in DD?
- How is f_{coupl} determined? What value is chosen here, why? What is its influence on the results?
- The velocity of dislocation Ia measured with TEM is given but not the velocity of dislocation II. The results from DD show that the two dislocations reach the same speed, is it the same with experiments? This should appear in the paper.
- The authors find that the nearest distance between the 2 gliding dislocations is too short to be measured at 200K, but that is not commented. Does it suggest that the nearest distance that enables gliding increases with temperature? Can it be related to the fact that the CRSS is lower at lower T (so shortest distance is needed to achieve larger stresses on dislocation II?)

Comments on the figures:

- Fig. 1b: there is a loop on the red curve around the point of coordinates (7.5,1) that is not commented anywhere. What does it stand for?
- Fig. 2b: why are the two stress curves not symmetric with respect to the zero relative position mark, and why are their maximums slightly shifted compared that same mark?
- Fig. 3i is hard to read and the resolution is not as good as the other figures. Please homogenize the color code between fig. 1a and fig. 3i for the 2 slip systems.
- Fig. 4 the fact that both [100] Beardmore and [100] CP (std) are displayed with lines is confusing and shades the improvement provided by [100] CP (coupl). The differences in line widths do not strike the eye. Maybe only put dotted lines (or no lines) for all simulations and plain lines for experiments?

Some remarks on syntax/typos:

- Dislocation Ia is not introduced in the text.
- Additional) between] and , on line 3 of paragraph 2, p. 5
- I don't understand the sentence "No slowing down in the in-situ TEM experiments." on p. 10

Dear Reviewers,

We thank the referees for their valuable and constructive comments and requests to clarify and improve the paper. In the following all points raised by the referees will be addressed. The changes in the paper are highlighted.

Reviewer #1 (Remarks to the Author):

The manuscript reports on a comparison between dislocation dynamics simulations and in-situ transmission electron microscopy regarding the repulsive interaction between two non-parallel screw dislocations. It is shown that the dislocations can glide together, producing an extra plastic deformation than can be linked to a lower strain hardening rate observed during deformation of bcc samples.

The work is potentially interesting although this Referee is concerned with the effect of the boundary conditions. In both DD and TEM, the dislocation ends are free, which allow the dislocations to glide over large distances. However, in a bulk sample, the dislocation ends would be part of a network and would be fixed, impeding the motion of the dislocations and limiting the production of plastic deformation. The same would be true in FCC metals: in absence of constraints on the dislocation ends, two repulsive dislocations can glide together and produce plastic strain. But this does not occur because dislocation ends are part of a network and cannot glide freely. Because of the nodes, the dislocation cannot remain straight and cannot maintain a constant velocity, in contrast with the present simulations and statements on P. 7, 12.

Answer:

The referee is right in pointing out the differences of a screw dislocation within a network and isolated screw dislocations as studied here.

In BCC metals, it is well-known that anchoring screw dislocations at their extremities do not stop the nucleation of kink-pairs and the dislocation motion (Caillard D., Acta Materialia 61 (2013) 2808–27, on the basis of the paper of Louchet F, Kubin LP, Vesely D. Philos Mag 1979;39:433). For the chosen material tungsten, experiments at room temperature are well within the low temperature regime and no complex dislocation networks but long screw dislocations dominate the micro-structure within the crystal.

In case of an anchored screw dislocation, the kink pair mechanism will still trigger its glide motion. At the network points (pinning points), a dislocation of mixed type would “accumulate” and finally glide through the sample or overcome the obstacles P1 and P2 as shown in figure 1 shown for the driven dislocation II.

The same holds for the driving dislocation I (light green). We agree that the velocity of the driving dislocation may be influenced by the network as the effective length changes, but not the one of driven dislocation as here the kink pair nucleation rate is controlled by the interaction zone only.

Therefore, the authors consider that studying the motion of a straight screw dislocation in a thin foil and in a thin layer (with or without periodic conditions) is representative for the elementary coupling mechanism proposed. The coupling scenario in both DDD and TEM is certainly idealized, but the main observation, that an initially immobile screw dislocation (dislocation II) starts to glide once dislocation I is close enough holds even if this is not as "smooth" in the experiments/reality as in the DDD simulations. In the DDD simulations, we suppress kink pair nucleation triggered by free surfaces (image forces).

The referees concern about the behavior in fcc metals is to the authors observation less problematic (or even irrelevant) for fcc metals as only a very small critical stress has to be overcome for dislocation glide. Due to the flexibility of fcc type dislocations, it is never observed in ddd simulations that interacting dislocation remain straight but often the dislocation twist and may even find an attractive orientation. In case of a repulsive orientation of dislocations in fcc (or with small critical stresses), which glide globally into the same direction, there will be a bowout in the glide direction along the driven dislocation, whereas an "inverse" bowout occurs on the driving dislocation and the net additional swept area of both dislocation cancel out approximately. Therefore the dislocation coupling will not (is very unlikely to) occur in fcc metals.

Other more minor concerns and comments:

- P.2 end of 1st paragraph: the referee does not understand « gives a formed component its strength » Isn't there a problem of English here?

Answer:

This is indeed an English problem: No the sentence reads as:

“This in turn is believed to result in work hardening which manifests itself as an increase in the flow stress of the material upon straining and gives e.g. a deep-drawn component its strength but also hinders further deformation.”

- P. 3 2nd paragraph: at which temperature were the [110] tractions performed?

Answer:

We added the temperature information to the paper. The tensile direction is along [100].

“At room temperature, for [100] tensile loading,...”

- P. 4: the referee does not agree that dislocation cores in bcc metals are extended and non-planar. It is quite the opposite. As shown by DFT calculations, non-degenerate core are highly compact. Having a compact core (and not an extended or non-planar core) is consistent with a high Peierls stress since it is known from the Peierls Nabarro model that compactness increases the Peierls stress. The authors should revise the corresponding paragraph

Answer:

The referee is right in pointing out, that “extended” is an old description for screw dislocation cores in bcc metals. On page 4: “extended” has been replaced by “complex”. A sentence has been added: “The core structure is found to be non-degenerate and compact. “

For the current paper, only the consequence being a high and stress dependent activation barrier is relevant, which was obtained from atomistic simulations.

- One of the authors has published in-situ observations in tungsten with very surprising dislocation behaviors. Has any of this peculiarities of tungsten been accounted for in the present study?

Answer:

The study mentioned here by the referee shows a strange behavior, with long jumps of screws in {112} planes, but only for a tensile axis [110]. For other tensile axes, the motion is a mixture of steady motion and small jumps in {110}, namely rather close to what is expected in case of the kink-pair mechanism. Here, in situ observations have been carried out with a tensile axis [521] and the second behavior is observed. It is thus not necessary to include the peculiar behavior restricted to straining along [110]. This phenomenon is neither addressed by the TEM or DDD results presented here.

Reviewer #2 (Remarks to the Author):

Authors report on a novel mechanism of dislocation interaction that acts contrary to conventional understandings of dislocation hardening in other metals. The observation is specifically recorded for tungsten-based metals, but shows broad generality to other BCC metals and systems that experience dislocation motion that relies on kink-pair nucleation processes. This particular observation is useful

in understanding enhanced ductility in heavily cold-worked BCC metals at temperatures that are generally considered to be below the ductile to brittle transition; these specific issues have not been adequately described in the literature to date.

The author's claims are well supported by experimental transmission electron microscopy (TEM) observations collected during time-resolved straining experiments, and modelling efforts support these findings with atomistically-informed discrete dislocation dynamics (DDD) simulations. Each of these methods has fundamental physical limitations with respect to time resolution and appropriate boundary conditions; however, the authors adequately address these concerns and provide a compelling argument for a new mechanism governing the coordinated motion of dislocations in this material.

There are a couple of minor issues that should be discussed prior to publication. First, why are the orientations between dislocations and loading directions (and the resulting Schmid factors) for driving and driven dislocations different between TEM and DDD modeling experiments?

Answer:

The setup was initially chosen to have numerically clear answer due to the very different Schmid factors on the driving and driven dislocations. The numerical evaluation of the kink pair nucleation critically depends on the summation of the nucleation probabilities which is done in a segment wise manner.

⇒ See also responses to 3rd referee

Second, why is the crystal plasticity modelling compared to the experimental results of Beardmore and Hull (1965) instead of the more recent publications by Brunner (2008, 2010), especially when the TEM experiments seem to be investigating material sourced from these latter investigations?

Answer:

The investigation of Brunner 2008 and 2010 were for a single slip orientation $[\bar{1}49]$ at different temperatures, showing also activity on other than the primary slip system. In the current investigation, we focused on multislip orientations to illustrate the effect of the coupled motion. Therefore we compare to the "old" results of Beardmore and Hull (1965).

Overall, the manuscript is of very high quality with strong and conclusive support of the author's claims. The impact of this research is expected to be broad and substantial within the field of materials science. Furthermore, the methods employed may be helpful for elucidating mechanisms of deformation behavior in a variety of novel materials that are used in extreme and demanding engineering environments.

P.S. Minor issues with figures and formatting:

Figure 1a) Difficult to tell orientation of n2. It appears to be orthogonal to n1 instead of normal to the plane containing Dislocation II. Figure 1 b) It would be helpful to clarify initial vs final configurations of dislocations and state that the critical distance is a function of the difference in Schmid factors (note: this is explicitly stated in a latter part of the manuscript). Also, what is the red circle at the intersection between dislocation 1 and 2? It looks like a continuation of the line describing dislocation II, but this mark and the truncation of the curves beyond the critical distance is not

addressed.

Answer:

Fig 1a) is improved and colors were homogenized (request by referee 3) and figure 1b the text to figure 1b has been clarified.

Figure S4.1) This figure appears to be unformatted, making it difficult to tell orientation of different vectors. I would recommend a 3D view of this figure in the final supplemental section to improve clarity. Also, the legend uses shorthand text instead of the specific variables identified in the supplemental manuscript.

Answer:

S4.1 has been improved

Reviewer #3 (Remarks to the Author):

This work uses a combination of multiscale modeling (DD and CP) with TEM observations of screw dislocations in W to evidence a new type of dislocation-dislocation interaction and its impact on work hardening in bcc metals. The results are new and of interest to the dislocation plasticity community as well as to a broader audience in the field of metallurgy and mechanical properties of materials.

Point by point comments:

- The title suggests the study concerns all bcc metals but the study was performed in W only. The reviewer agrees that the mechanisms evidenced here are most likely transferable to other bcc metals, but the justification for the use of W is not clear.

Answer:

The choice of tungsten as model materials was done for several reasons: The material is elastically isotropic, which makes it more easily accessible for discrete dislocations dynamics. For tungsten, experiments at room temperature are well within the low temperature regime and no complex dislocation networks but long screw dislocation dominate the micro-structure within the crystal. Furthermore the atomistic information on the stress dependence of the activation barrier were available for tungsten and not similarly complete available for other bcc materials.

The coupling phenomenon has also been observed for Fe and Nb at $T=95K$ and corresponding TEM information has been added to the supplement S2 (Figures S2.2 S2.3).

- The abstract does not follow standard rules (it starts with a result for example) and is too technical for a broad audience. The introduction however is very pedagogic so the reviewer believes the authors can improve the abstract.

Answer:

The first part of the abstract has been reformulated to improve readability.

- The text mentions several times that the DD and CP models are atomistically informed but remains vague as to how: which atomistic models are used as input, what phenomenon are taken into

account (the fact that non-Schmid effects are accounted for is not mentioned in the main text, only in the Supplementary files), and how that may influence the results obtained here, especially the Schmid factors:

Answer:

Concerning the DDD simulations: the fundamental mechanism of coupling does not depend on the details of atomistic parameters, only the stresses or critical distances do. However, it is crucial to have atomistic information on the stress dependence of the activation barrier. Therefore the authors refrained from detailing the atomistic aspects in the main paper. This has been published before and is referred to.

On page 4 the inclusion of non-Schmid effects is now explicitly mentioned.

- The parameters given in S5 for the CP calculations do not include T/AT asymmetry ($a_1 = 0$)

Answer:

For the CP model, an “effective resolved shear stress” is computed including all stress components required to parametrize the atomistic results of Gröger 2008 [ref19 of paper]. Here it turned out that T/AT is absent in case of tungsten, reflected by setting $a_1=0$.

- The authors find a different value of distance between gliding dislocations with DD and with TEM and link it to the different Schmid factors. Could a quantitative link be made there? Were non-Schmid effects taken into account when calculating the Schmid factors with DD?

Answer:

The Schmid factor given in the paper do not include / mimic non-Schmid effects: The dependence of stress components other than the resolved shear stress is included in the activation enthalpies used for the kink pair nucleation rate. Thus the complete stress dependency is taken into account.

The question is related to the different loading orientations for DD and TEM. First, activation energies for screw dislocations obtained from atomistic, are known to lead to higher stresses compared to experimental observations and a quantitative agreement is therefore not expected. Second, the role of the loading directions and thus the different Schmid factors on dislocation I and II on the critical distance can well be quantitatively understood: For the chosen setup in DD $[\bar{1}49]$, the Schmid factor ratio is $\frac{0.5}{0.3} \approx 1.67$ and the critical distance is about 7.3nm (23 a; a lattice spacings). Using the orientation [521] from the experiments (TEM) the ratio is $\frac{0.485}{0.458} \approx 1.056$ and the corresponding critical distance in DD is about 32a. This leads to a ratio of the critical distances for the [521] and [149] direction of $\frac{32}{23} \approx 1.39$. The corresponding interaction stress on the driven dislocation increases by the same ratio and thus compensates the difference between the differences in the Schmid factor of the dislocation I and II in the respective directions, which amount to $0.61 = 0.67 - 0.056$. The smaller Schmid factor on dislocation II is compensated by larger interaction stresses. It is not

possible to give a simple relation between the critical distance ratio and Schmid factor differences as the activation enthalpies depend in case of tungsten on two components of the stress tensor.

- Can non-Schmid effect be a source of discrepancy between the results from CP and experiments for [111] loading in Fig.4?

Answer:

The non-Schmid effects are very pronounced for the [111] direction. Without non-Schmid effect the [111] curve would be roughly 30% higher and thus way off the experimental "saturation" values. Also the stress-velocity relation (power law) and its exponent has some influence on the precise shape. If one assumes that the flow criterion (effective flow stress; Gröger 2008 [ref 19 of paper]) is correct, one comes to the conclusion that the hardening law and stress-velocity relation must be improved, to achieve a better agreement with experiments.

- Why was $\bar{\epsilon}_{49}$ chosen for the tensile loading axis in DD?

Answer:

We aimed at a clear distinction between the different driving factors and have therefore chosen this setup. The numerical evaluation of the kink pair nucleation critically depends on the summation of the nucleation probabilities which is done in a segment wise manner.

- How is f_{coupl} determined? What value is chosen here, why? What is its influence on the results?

Answer:

The value of f_{coupl} was chosen to be 0.5 (in the middle of the interval which "seems" reasonable) to illustrate the effect. Physically a value of 1 would mean, that each driving dislocation triggers a "repulsive" event. For a value of 0.5 every other dislocation triggers such an event. The coupling factor directly scales the additional plastic slip contribution. As this additional contribution to the plastic slip does not lead to work-hardening, the flow stress is always lower compared to the one of standard CP for the same total strain.

- The velocity of dislocation Ia measured with TEM is given but not the velocity of dislocation II. The results from DD show that the two dislocations reach the same speed, is it the same with experiments? This should appear in the paper.

Answer:

The behavior of the velocities is quite similar in TEM: the velocity of dislocation II varies around the value of dislocation 1a respective 1b but it is not as smooth.

A corresponding sentence has been added to the paper on page 9.

- The authors find that the nearest distance between the 2 gliding dislocations is too short to be measured at 200K, but that is not commented. Does it suggest that the nearest distance that enables gliding increases with temperature? Can it be related to the fact that the CRSS is lower at lower T (so shortest distance is needed to achieve larger stresses on dislocation II?)

Answer:

Yes, a lower temperature requires larger driving stresses, in particular a larger interaction stress and thus a smaller critical distance.

Comments on the figures:

- Fig. 1b: there is a loop on the red curve around the point of coordinates (7.5,1) that is not commented anywhere. What does it stand for?

Answer:

The red curve "circles" around a point as the driven dislocation has a velocity variation. On average both dislocations glide with the same velocity. As the motion occurs by discrete kinks in the respective glide directions of the dislocations (jump length) the minimal distance also varies (horizontal axis). As the velocity of dislocation I is controlled by the kink pair nucleation rate along the entire dislocation length, no velocity fluctuation is visible for dislocation I. In case of dislocation II, the velocity is controlled by the zone of high interaction stress and thus small variations in the minimal distance lead to some velocity fluctuations.

- Fig. 2b: why are the two stress curves not symmetric with respect to the zero relative position mark, and why are their maximums slightly shifted compared that same mark?

Answer:

The figure shows the resolved shear stress in the glide planes of the respective dislocations. As the referee noted, the total norm of the Peach-Koehler force is symmetric with respect to the point of nearest approach. The symmetry is broken by projecting the total Peach-Koehler force on the glide planes of the dislocations (resolved shear stress). This leads to both an offset of the maximum and an asymmetry with respect to the relative position along the dislocation(s).

The following sentence has been added on page 7: "The curves are asymmetric with respect to the nearest point of interaction as the respective resolved shear stresses are shown."

- Fig. 3i is hard to read and the resolution is not as good as the other figures. Please homogenize the color code between fig. 1a and fig. 3i for the 2 slip systems.

Answer:

The figure has been improved accordingly.

- Fig. 4 the fact that both [100] Beardmore and [100] CP (std) are displayed with lines is confusing and shades the improvement provided by [100] CP (coupl). The differences in line widths do not strike

the eye. Maybe only put dotted lines (or no lines) for all simulations and plain lines for experiments?

Answer:

The line type and symbols have been adapted so that a clear assignment of the lines is possible when printing in grayscale. Line and dashed line without symbols are used for the experimental curves. For all simulations short dashed lines are used for the CP(std) with symbol and dotted lines with symbol for CP(coupl).

Some remarks on syntax/typos:

- Dislocation Ia is not introduced in the text.

Answer:

The following sentence has been added to clarify the labelling of dislocation Ia and Ib on page 8.

"In the TEM sample, the first respectively second driving dislocation is labelled dislocation Ia respectively dislocation Ib."

- Additional) between] and , on line 3 of paragraph 2, p. 5

Answer:

The ")" has been removed.

- I don't understand the sentence "No slowing down in the in-situ TEM experiments." on p. 10

Answer:

The sentence has been removed.

Reviewers' Comments:

Reviewer #1:

Remarks to the Author:

The Referee is not satisfied with the corrections made by the authors to the manuscript. They have only accounted for the minor remarks, but have answered the main concern about boundary condition effects. The authors communicated an answer but did not revise the manuscript accordingly.

Moreover, the answer given is not convincing. Obviously, anchoring the dislocations will not impede kink-pair nucleation but the accumulation of kinks at the network nodes will increase the line length and thus will produce a restoring force that will not only affect the velocity, but can stop the dislocation and thus block the mechanism reported in the manuscript in bulk samples.

The Referee is confident that many readers will have the same concern, which should therefore be treated seriously by the authors. They should perform simulations with nodes to make sure that the same mechanism can be activated in a network configuration.

Reviewer #2:

Remarks to the Author:

The authors are comprehensive in addressing the questions and requests for clarification stated in the original review.

Reviewer #3:

Remarks to the Author:

This referee thanks the authors for their responses.

However, some points discussed in the review such as the choice of value for f_{coupl} and the link between critical distance and temperatures were not added in the text.

Other than this the paper was edited according to the reviewers' comments and the authors' responses.

Dear Reviewers,

We thank again the referees for their valuable and constructive comments and requests to clarify and improve the paper. In the following all points raised by the referees for the revised paper will be addressed. The changes in the paper are highlighted.

Reviewer #1 (Remarks to the Author):

The Referee is not satisfied with the corrections made by the authors to the manuscript. They have only accounted for the minor remarks, but have answered the main concern about boundary condition effects. The authors communicated an answer but did not revise the manuscript accordingly.

Moreover, the answer given is not convincing. Obviously, anchoring the dislocations will not impede kink-pair nucleation but the accumulation of kinks at the network nodes will increase the line length and thus will produce a restoring force that will not only affect the velocity, but can stop the dislocation and thus block the mechanism reported in the manuscript in bulk samples.

The Referee is confident that many readers will have the same concern, which should therefore be treated seriously by the authors. They should perform simulations with nodes to make sure that the same mechanism can be activated in a network configuration.

Answer:

We have taken up the reviewer's suggestion and have now included a revised response in the discussion section of the manuscript to address the critical points about the role of dislocation length extension due to network anchor points on screw dislocation motion. The authors did not follow the suggestion to include additional simulation results, but included a detailed answer referring to the different physical interactions occurring within a network. The following paragraph has been added on page 11:

“ The in-situ observations and DDD results are both obtained for straight dislocations, reaching from surface to surface. In bulk material, dislocations are often anchored in networks which leads to an increase in line length during motion. One could speculate that this might possibly suppress screw dislocation motion and hence the coupling mechanism. However, the increase of length corresponds to the nucleation of new kink pairs, which takes place anyway, even in case of straight screws ending at the surfaces. These kinks subsequently either accumulate at the dislocation extremities in the case of bulk material or disappear at the surfaces. This increase of length is actually a restoring force, but already included in the kink-pair nucleation process. Therefore, the only ingredient, which has not been taken into account for mimicking bulk behavior, is the elastic repulsion between kinks accumulated at the screw extremities – forming a mixed dislocation (curved section with non-screw character). This repulsion is part of the internal stress due to elastic interactions with all neighboring dislocations, which of course must be taken into account for kink-pair nucleation in bulk. However, this is very different from the restoring forces due to line tension effects observed in fcc metals. “

Reviewer #2 (Remarks to the Author):

The authors are comprehensive in addressing the questions and requests for clarification stated in the original review.

Answer:

We thank the reviewer for reading the revised version of the paper.

Reviewer #3 (Remarks to the Author):

This referee thanks the authors for their responses. However, some points discussed in the review such as the choice of value for f_{coupl} and the link between critical distance and temperatures were not added in the text.

Answer:

The reasoning behind choice of f_{coupl} has been added in brief in the main paper on page 15 and in more detail included in Supplement S5.

The following sentence explaining the link between the critical distance and temperature has been added to the main paper on page 11 at the end of second paragraph.

“The smaller critical distance between the coupled pairs observed leads to a larger interaction stress, as required for activating screw dislocation motion at lower temperatures”.

Other than this the paper was edited according to the reviewers' comments and the authors' responses.

Reviewers' Comments:

Reviewer #3:

None